# *Chondracanthus teedei* var. *lusitanicus*: The Nutraceutical Potential of an Unexploited Marine Resource

**DOI:** 10.3390/md19100570

**Published:** 2021-10-14

**Authors:** Diana Pacheco, João Cotas, Andreia Domingues, Sandrine Ressurreição, Kiril Bahcevandziev, Leonel Pereira

**Affiliations:** 1MARE—Marine and Environmental Sciences Centre, Department of Life Sciences, University of Coimbra, Calçada Martim de Freitas, 3000-456 Coimbra, Portugal; diana.pacheco@uc.pt (D.P.); jcotas@uc.pt (J.C.); 2Polytechnic of Coimbra, Coimbra Agriculture School, Bencanta, 3045-601 Coimbra, Portugal; 21829002@esac.pt (A.D.); sandrine@esac.pt (S.R.); kiril@esac.pt (K.B.); 3Research Center for Natural Resources, Environment and Society (CERNAS), Applied Research Institute (I2A), Coimbra Agriculture School, Bencanta, 3045-601 Coimbra, Portugal

**Keywords:** red seaweed, nutritional characterization, nutraceutical, bioactive compounds

## Abstract

Presently, there is a high demand for nutritionally enhanced foods, so it is a current challenge to look at new raw food sources that can supplement beneficially the human diet. The nutritional profile and key secondary metabolites of red seaweeds (Rhodophyta) are gaining interest because of this challenge. In this context, the possible use of the red seaweed *Chondracanthus teedei* var. *lusitanicus* (Gigartinales) as a novel nutraceutical source was investigated. As a result, we highlight the high mineral content of this seaweed, representing 29.35 g 100 g^−1^ of its dry weight (DW). Despite the low levels of calcium and phosphorus (0.26 and 0.20 g 100 g^−1^ DW, respectively), this seaweed is an interesting source of nitrogen and potassium (2.13 and 2.29 g^−1^ DW, accordingly). Furthermore, the high content of carbohydrates (56.03 g 100 g^−1^ DW), which acts as dietary fibers, confers a low caloric content of this raw food source. Thus, this study demonstrates that *C*. *teedei* var. *lusitanicus* is in fact an unexploited potential resource with the capability to provide key minerals to the human diet with promising nutraceutical properties.

## 1. Introduction

The world’s population is rapidly increasing, placing pressure on traditional food sources, and causing adverse impacts all over the world. As a result, there is a need to discover new raw food sources that can provide vital nutrients and minerals to humans to aid body cell function [1]. Due to the overexploitation of arable land, seaweed has potential for food supply, as it does not compete with terrestrial plants, and can be produced in different systems such as in depleted salt pans, estuarine water, or offshore farming systems [2].

There are several records that show that seaweed has been incorporated in the daily diet of coastal populations and cultures in Europe, Japan, and China since ancient times [3,4,5,6]. Currently, Europe consumes nearly 97 tons of seaweeds each year, of which the majority are imported [7].

Allied to the need to provide novel nutritious and healthy food sources to ensure global food security, seaweeds are being explored to be included in a list of possible food sources for animal and human diets [8,9].

Red seaweeds are recognized as a possible source of several unique metabolites with a variety of health benefits, in addition to their considerable contribution to the nutritional and industrial supplies [10]. For example, algal polysaccharides contribute as low caloric food sources, acting as anti-obesity agents, since these compounds are not digested by the human organism and provides the sense of satiety [11]. Furthermore, seaweeds synthesize essential fatty acids and essential amino acids, which are only acquired through the daily diet [12]. Because macroalgae are so rich in a variety of minerals that are needed for human health, red seaweeds are seen as valuable resources to be used as nutraceuticals [13,14]. Structural functions, such as tissue and bone synthesis, as well as regulatory functions, such as enzymatic activity, oxygen transport, and neurotransmission are supported by minerals acquired through the daily diet [12,13,14,15]. In fact, chronic diseases, such as obesity, diabetes, cardiovascular diseases, and the increased risk of certain types of cancers are related to consumption patterns and dietary factors [16]. Mineral elements are connected with organic substances, so the lack in one mineral will have an impact on the functioning of others [17]. For this reason, adequate mineral intake is needed to maintain the proper functioning of the human organism and cell homeostasis [12,16]. Minerals, such as calcium, magnesium, phosphorus, sodium, and potassium are required in larger quantities than trace minerals, such as iron, zinc, or copper [17]. Nevertheless, the ingestion of these trace elements is equally important to other minerals [18]. A favorable balance of mineral elements is required for the body to generate new tissue during growth, pregnancy, lactation, and other specific stress situations [17]. Moreover, the requirements of minerals intake vary according to the age and gender [17,18,19]. For instance, women during pregnancy are more likely to have low iron and calcium levels because their food consumption does not always match their needs, putting them at risk of developing anemia or osteoporosis, respectively [16]. Furthermore, due to poor-quality diets, diseases and therapies with an impact on a nutrient absorption or use, many people, particularly the senior population, are exposed to mineral deficiencies [18].

However, more crucial than emphasizing the health benefits of consuming seaweeds, is ensuring its long-term production through aquaculture while preserving its nutritional features [1,2]. Novel food sources, such as seaweeds, do not compete with ecological niches (sustainable food source), must present an important nutritional content in order to be used as nutraceutical output in the human diet [1], to reduce disease incidence, such as cancer, cardiovascular-associated diseases and problems related with undernourishment [1]. However, in the seas there are also wild and unstudied seaweeds that could be harmful for human health, when consumed, due to their retention of heavy metals, toxins and noxious chemicals, leading to the need for monitoring their production and/or harvesting [2,14].

One of the lesser known seaweeds, *Chondracanthus teedei*, has already proved to be a viable food source, and a well-adapted species that can be produced through aquaculture [20,21], showing that may be grown with consistency in terms of nutrients, assuring as a safe food product and, decreasing the risk of dangerous compounds ingestion [1,2].

The red seaweed *Chondracanthus teedei* var. *lusitanicus* (Gigartinales), which inhabits the Portuguese sea, is a variation of the species *Chondracanthus teedei* that can be found in the Atlantic Ocean, Mediterranean and Black Sea [22]. *C. teedei* var. *lusitanicus* (J.E. De Mesquita Rodrigues) Bárbara and Cremades 1996 (Figure 1) is a cosmopolitan species typically present on lower intertidal and shallow subtidal habitats, can be found in semi-exposed or protected areas, tolerates mud and/or sand [23]. Its biochemical, but not nutritional, composition varies depending on life cycle stage, geographical location, and the harvesting season [24].

It is highlighted in the literature that the nutritional potential of *C. teedei* in the human diet [1] but less known in *C. teedei* var. *lusitanicus*. *C. teedei* var. *lusitanicus* has already shown some fascinating properties, synthesizing sulfated polysaccharides (carrageenans) with antifungal activity [24], considered to be, potentially, an edible seaweed well-suited for consumption either fresh or processed [25] and sustainably produced by aquaculture [26]. For this reason, the aim of this study was to assess the nutritional profile of *C. teedei* var. *lusitanicus*, collected from its natural habitat, and to evaluate whether it has the potential to be a nutraceutical food source for humans.

## 2. Results

### 2.1. Macro- and Micro-Element Profile

The red seaweed *C. teedei* var. *lusitanicus* has shown an interesting macro- and micro- element profile (Table 1), exhibiting a high content in nitrogen and potassium (2.13 and 2.29 g 100 g^−1^, respectively), comparatively with the content of copper, zinc and manganese (3.0 × 10^−4^, 2.4 × 10^−3^ and 1.2 × 10^−3^ g 100 g^−1^, accordingly), which were the lowest values in the elemental analysis of this seaweed biomass.

### 2.2. Nutritional Evaluation

When the nutritional evaluation (Table 2) of the fresh (FW) and dried (DW) *C. teedei* var. *lusitanicus* biomass weight was analyzed, it was observed that the DW is more encouraging from a nutritional aspect. A major part of this seaweed is composed of water. Meanwhile, it is highlighted that among the nutritional components analyzed that both FW and DW, contain a high content of crude carbohydrate presenting respectively 7.55 and 56.03 g 100 g^−1^. FW and DW also showed to be a significant source of protein (1.54 and 11.42 g 100 g^−1^) and minerals (3.96 and 29.35 g 100 g^−1^). It was also observed that the ratio DW:FW of *C. teedei* var. *lusitanicus* is 1:7, for all nutritional parameters, meaning that 1 g of dried algal biomass corresponds, nutritionally, to 7.4 g of fresh seaweed.

### 2.3. Polyssacharides from C. teedei var. lusitanicus

Carbohydrates are the most abundant nutrient in *C*. *teedei* var. *lusitanicus (*Table 3), emphasizing the polysaccharide content of this seaweed. When compared all samples, the female and male gametophytes of *C. teedei* var. *lusitanicus* produced the highest polysaccharide quantity, 40.9 and 42.1%, respectively.

#### 2.3.1. FTIR-ATR

The isolated polysaccharides were examined using FTIR-ATR. This spectroscopic approach enabled quick, nondestructive polysaccharide characterization with a small amount of sample [28]. The bibliography was used to assist the collection of the spectra [29,30].

Along with the presence of three shoulder peaks at 905 cm^−1^, 930 cm^−1^, and 1070 cm^−1^ in the FTIR spectrum, which is associated with the presence of theta-carrageenan, the *C*. *teedei* var. *lusitanicus* tetrasporophyte (Figure 2 and Table 4) has a hybrid xi/theta carrageenan [25,29]. Moreover, the tetrasporophyte phase of *C*. *teedei* var. *lusitanicus*, has a large peak in 830 cm^−1^, which is typical of two principal peaks near the xi-carrageenan. The large and prominent peak in this case reveals that this life cycle phase of *C*. *teedei* var. *lusitanicus* tetrasporophyte (Figure 3A) possesses also an xi/theta carrageenan [25]. The FTIR-ATR spectra of male and female *C*. *teedei* var. *lusitanicus* gametophytes (Figure 2B,C) were identical, indicating the existence of a hybrid kappa/iota carrageenan (present in the peaks: kappa: 930 and 845 cm^−1^; iota: 805 cm^−1^).

#### 2.3.2. ^1^H-RMN

The anomeric protons zone of the ^1^H-NMR (Figure 3) spectra of native and alkali-modified carrageenan’s from *C*. *teedei* var. *lusitanicus* female and male gametophytes revealed two strong signals at 5.11 ppm and 5.32 ppm, respectively. The anomeric protons of 3,6-anhydro-α-d-galactose (DA; kappa-carrageenan) and 2-sulfated 3,6-anhydro-α-d-galactose (DA2S; iota-carrageenan) are represented by these signals. A weaker signal with a chemical shift of 5.37 ppm was detected in the spectrum of the alkaline-extracted female gametophyte sample, in addition to the carrageenan signals. This signal has been attributed to floridean starch, a common and natural contaminant found in carrageenan samples [32,33,34,35].

#### 2.3.3. ^13^C-RMN

The anomeric region of the ^13^C-NMR spectra of alkaline-extracted carrageenan (female and male gametophytes) contains three significant peaks (Figure 4B,C): 102.5 ppm corresponds to anomeric carbon of β-d-galactose-4-sulfate residues (G4S) found in both kappa- and iota carrageenans; 95.3 ppm corresponds to anomeric carbon of 3-6-anhydro-galactose (DA) in kappa-carrageenan; and 92.1 ppm corresponds to anomeric carbon of anhydro-galactose-2-sulfate (DA2S) [32,34,35,36].

Figure 4T show signals at 103.3 and 92.8 ppm, 100.4 and 95.7 ppm that might be ascribed to the anomeric carbons of xi and theta-carrageenan, respectively, in the ^13^C-NMR spectra of native and alkaline-extracted carrageenan from tetrasporophytes [37,38].

## 3. Discussion

Among the algal phyla or classes, Chlorophyta (green), Phaeophyceae (brown) and Rhodophyta (red), the red seaweeds phyla contain the largest number of species [23]. Hereby, it is highlighted the nutraceutical potential that this phylum/class represents [39]. Seaweeds have been seen as a feedstock for bioactive molecules that can be incorporated in the daily diet as a supplement in order to promote human health, thus being considered nutraceutical food products [40].

The incorporation of the red seaweed *Chondracanthus teedei* in the daily diet has been recorded in some coastal areas of Southern Europe [41]. In some other parts of the globe, *C. teedei* revealed to be a potential food supplement with nutraceutical potential: so why not *C. teedei* var. *lusitanicus*? For instance, in Brazil, the dried biomass of *C. teedei* (at 45 °C for 48 h) revealed to be constituted by 14.66% of protein, 2.21% of fibers, 1.82% of total lipids, a mineral content of 28.68% and a moisture content of 86.73% [42]. A study conducted with *C. teedei* were, in overall, higher than those recorded in the current research for *C. teedei* var. *lusitanicus*, which may be attributable to differences in geographical harvesting sites, different physical-chemical parameters of the seawater, abiotic and biotic factors interaction, or even differences in seaweed processing, such as the drying process [43,44,45].

Previous findings already showed that the biochemical characterization of *C. teedei* can vary within the geographical area and the harvest season [46], particularly the nitrogen content, which suffers a decrease during the summer and autumn [21]. Indeed, several factors can influence nutrient uptake (particularly nitrogen and phosphorus), such as light, temperature, hydrodynamics, desiccation, and salinity [47,48]. In fact, researchers found that *C. teedei* var. *lusitanicus* harvested at Cabo Mondego (Figueira da Foz) in the summer of 2019 showed a lower protein content (0.2 g 100 g^−1^ DW) [49], which may be due to the different seaweed processing, particularly the drying process [50]. Because seaweeds have a low lipid content, they are considered a low-fat food [51]. Furthermore, it was discovered that the lipid profile of this seaweed can change substantially in terms of both qualitative and quantitative variation depending on its life cycle phase [52,53].

Within the threshold set by the competent authorities, the daily reference intakes of minerals and trace elements established by the European Parliament and the Council of European Union, as well as the recommended dietary allowances and adequate intakes established by the United States of America, *C. teedei* var. *lusitanicus* represents an adequate mineral source (namely of nitrogen, phosphorus, calcium, and magnesium) [54,55]. However, according to these results, a 100 g portion of this seaweed can in fact exceed the recommended dietary allowance and the adequate intake trace elements, particularly iron (0.014 g—EU; 0.008 g—USA), copper (0.001 g—EU; 0.0009 g—USA), zinc (0.01 g—EU; 0.011 g—USA) and manganese (0.002 g—EU; 0.0023 g—USA) [54,55]. Thus, the recommended intake is about 7 g (DW) of this seaweed species per day, due to the high content of these micronutrients. Particularly the iron content, which only 7 g of DW seaweed are near 9.15% of the iron daily intake (Table 1). This is consistent with the literature, which states that the maximum amount of dried seaweeds that may be consumed with favorable effects for human health is roughly 7 g, after which the normal function of cell mechanisms can be directly impacted [14,56].

The supply of macrominerals, such as calcium, magnesium, phosphorus, sodium, and potassium are required in larger amounts and needed for the proper functioning of the human body [17].

For instance, calcium is essential for several vital functions, since this mineral is a structural component of bones and teeth, cell membrane, and it is involved in neuromuscular activity, endocrine secretory function, and blood coagulation [57]. For this reason, the lack of this mineral can be reflected on low bone density, which could lead to the development of osteoporosis [57].

The lack of phosphorus can lead to the deterioration of mental function and in severe cases the patients can develop osteomalacia [57,58]. This happens because phosphorus is involved on regulatory processes, such as energy metabolism, pH maintenance, glycolysis, muscle, and nerve function, but is also a structural element on bone and cell membrane synthesis [57]. While sodium, as an electrolyte, has a regulatory role on the human body, regulating the water distribution, and being involved on the active transport of molecules through the cellular membrane [59,60]. For this reason, a diet poor in sodium can lead to dehydration by reduction of the body fluid volume [59,60].

Additionally, magnesium is a key player for human cells as cofactor for more than 300 enzymes that regulates various biochemical processes in the human body, including protein synthesis, muscle and nerve function, blood glucose control, and blood pressure regulation [61,62]. Furthermore, 69 g of spinach or 105 g of kale contains the same magnesium content as this seaweed [63].

Potassium is one of the major intracellular cations, present in all human body cells because their inherent requirement for the cell homeostasis, as intracellular fluid regulator (osmosis process) and to intermediate the transmembrane electrochemical gradients (nervous signaling pathways and muscular function) [59,64]. Additionally, potassium helps to maintain the extracellular fluids, such as the plasma and blood [59,65]. This seaweed DRI is similar to 17.60 g of beet greens or 34.33 g of spinach [66].

As it shown in Table 1 and Table 2, this seaweed is also a rich supply of microminerals: iron, manganese, copper and zinc, as well as dietary fibers and protein, displaying a strong nutraceutical source of key elements and compounds for human welfare [1].

Only 7 g DW of *C*. *teedei* var. *lusitanicus* contains the same amount of iron as 226 g of broccoli or potato [67], this is significant since iron is required for the metabolism of many human proteins, including enzymes and hemoglobin (responsible for oxygen and carbon dioxide transportation system) [68]. Furthermore, disorders caused by a shortage of iron (e.g., anemia) can be alleviated by iron supplementation in the daily diet [69].

The manganese content of *C*. *teedei* var. *lusitanicus* is equivalent to 8.4 g sweet potato or 7.6 g brown rice [70]. Manganese is an important cofactor for enzymes such as manganese superoxide dismutase, arginase, and pyruvate carboxylase [71,72]. Furthermore, this mineral is involved in a variety of metabolic processes, such as amino acid, cholesterol, glucose, and carbohydrate metabolisms. Additionally, manganese is involved in the scavenging of reactive oxygen species, bone formation, reproduction, and immunological response [73,74,75].

Just 7 g of this seaweed contains the same amount of copper as 7.6 g of sweet potato or 0.95 g of sesame seeds [76], being a necessary micronutrient due to its involvement in numerous biological processes such as antioxidant defense, neuropeptide production and immunological function [77,78].

Zinc is a critical microelement that plays a regulatory and structural role in cell membrane stabilization, and only 7 g of this seaweed can provide the same amount of zinc as 12.9 g of Shiitake Mushrooms or 21 g of spinach [79].

Although some minerals are beneficial at low concentrations, they are extremely dangerous at medium concentrations, even causing human mortality. As a result, in order to attain the greatest advantage, this seaweed must be ingested in small doses, as indicated, making it a food supplement containing essential minerals with nutraceutical potential for human health [1].

The fiber level of 7 g of *C. teedei* var. *lusitanicus* is equivalent to 6 g of broccoli or 4 g of sweet potato [67]. The fiber in this seaweed is primarily carrageenan [25], and the key benefits are that it is digestible and swells in aqueous solution. As a result, carrageenan increases satiety and weight loss by decreasing stomach discharge and so providing improved glycemic control (reducing hyperglycemia-related illnesses). Furthermore, fibers have a favorable impact on the microbiome and gut transit in the gastrointestinal system [80].

Proteins are the fundamental building blocks of human cells, as well as the precursors of enzymes, antibodies, and hormones [81]. Hence, it is highlighted that the protein content in 7 g of the studied seaweed can replace 37 g of asparagus or 29 g of broccoli in a meal [67].

Carbohydrates are unique to each taxonomic group of seaweeds; and are also the major components of seaweeds biomass, which the human digestive system does not digest (acting as dietary fibers) [1]. However, seaweeds contain monosaccharides that despite having a low proportion, provide energy to human cells, to work and sustain their metabolism and regular functioning [1].

This nutritional comparative analysis highlights the quality of *C. teedei* var. *lusitanicus* as a nutritious food source, which has a potential to be used as a food supplement to enrich the deficiency of several macro and micronutrients in a typical diet. This seaweed can also be used as a food nutrient source for people that are living in developing countries as well as an alternative source of several minerals required for human cell homeostasis [82].

Important bioactive compounds that must be present in the human diet, such as proteins, lipids, carbohydrates, and minerals, are usually acquired through the vegetable consumption. Nevertheless, our results show that only 7 g (DW) or 50 g (FW) of *C. teedei* var. *lusitanicus* can contribute to a healthy diet than higher amounts of vegetables. Hence, it is highlighted the nutraceutical potential of this unexploited marine resource.

Although the nutritional and mineral profiles of this seaweed are consistent across all life cycles, the polysaccharide profile differs from gametophytes to the tetrasporophyte [22].

According to Pereira (2004) [83], the carrageenan extraction yield of *C. teedei* var. *lusitanicus* (female and male gametophyte, tetrasporophyte) is consistent, even the sampling site was the same in both studies.

Regarding the spectroscopic analysis, it revealed that the two phases of the life cycle of *C*. *teedei* var. *lusitanicus* showed similar variation, which was also found in other *Chondracanthus* genus species [84]: carrageenans of the kappa family are produced by the gametophyte stages (hybrid kappa/iota/mu/nu carrageenan), whereas carrageenans of the lambda family are produced by the tetrasporophyte stages (hybrid xi/theta-carrageenan).

The male gametophyte of *C*. *teedei* var. *lusitanicus* appears to possess a similar phycocolloid to the female gametophytes, based on the FTIR-ATR, ^1^H- and ^13^C-NMR spectra of its carrageenans. Even though both thalli formed a kappa/iota hybrid carrageenan (Table 1), the male thalli had equal levels of kappa and iota-fraction, but the kappa-fraction was slightly dominant in female gametophytes.

Regarding the male and female gametophytes alkaline-extracted carrageenan, our findings are similar to those of Pereira et al. (2004) and Zinoun et al. (1993) [32,85]. The existence of xi-carrageenan is indicated by the FTIR spectra of tetrasporic samples of *C*. *teedei*, which display stronger peaks at 830 cm^−1^, but less absorption at 820 cm^−1^. The presence of a hybrid xi/theta-carrageenan was confirmed by the ^13^C-NMR spectra, despite the ^1^H-NMR spectra not being conclusive (results not reported).

Following the minerals, polysaccharides, initially identified as carbohydrates, were found to contribute for a significant portion of the DW of the seaweed biomass. However, most of these polysaccharides are carrageenans, known as dietary fiber, due to its inability to break the high weight molecule in the human digestive tract.

Because of the negative effects that can occur if the cumulative dosage of seaweed polysaccharides (and particularly their lower-molecular-weight oligomers) exceeds the limit of 25 g/day, the diversity of seaweed polysaccharides (and particularly their lower-molecular-weight oligomers) needs to be quantified [86,87]. The recommended biomass intake of *C*. *teedei* var. *lusitanicus* ranges between 11 and 7% of this value, indicating that it is a supplement for human food intake that promotes human wellness.

Polysaccharides from seaweed with a high molecular weight are typically thought to be desirable dietary fibers. These are considered to be essential contributors in human health and illness prevention in certain applications [88]. These advantages are enlarged because the gut microbiome interacts with the host at both the intestinal and systemic levels, resulting in balance between the host and the microbiota. Food intake can have a beneficial or negative impact on the microflora balance, resulting in immunological, physiological, metabolic, and even psychological effects [88,89].

In addition to their biological properties, seaweed polysaccharides also have innate features that are very important for intestinal health; these include the viscosity and the high potential for water-binding activity, which adjusts the transit time of food through the gut. Such properties are demonstrated to promote satiety and weight loss; additionally, they delay gastric discharge, therefore promoting glycemic control (i.e., minimizing the incidence of diabetes). In the intestinal tract, all seaweed-derived polysaccharides are reported to enhance gut transit, maintaining regular stool volume, and promote beneficial alterations to the composition of the microbiome [14]. These advantages add up to improved metabolization of volatile fatty acids (VFAs), which are also known as short chain fatty acids (SCFAs), by members of the microflora, promoting positive effects in the gastrointestinal system, and thus improving cardiometabolic, immune, bone, and health conditions [14,90,91,92].

Due to the polysaccharides fermentation in the intestinal tract, reducing the microflora/bile salt hydrolase activity, anti-obesity effects have been identified as one of the most advantageous properties of seaweed polysaccharides for human ingestion, which is one theory for this observed effect [93,94,95]. In this case, in vitro experiments revealed that the microbiome composition changed to an enhanced condition, including populations of *Bifidobacterium*, *Bacteroides*, *Lactobacillus*, *Roseburia*, *Parasutterella*, *Fusicatenibacter*, *Coprococcus* and *Fecalibacterium* colonies [93,94,95].

Carrageenans are one of the most bioactive polysaccharides found in seaweeds; their chemical structure allows them to form hydrogels, allowing them to be employed as antiviral and antibacterial components in a myriad of product formulations [96,97]. Given the excellent levels of safety, efficacy, and biocompatibility, as well as the fact that they are biodegradable and non-toxic, there are strong reasons to use these compounds [98]. Carrageenan has also been used as a traditional medicine to treat coughs and the common cold, according to ancient documents, hence in vitro and in vivo experiments using animal models have corroborated this ethno-botanical information. The actions of carrageenans on blood platelet aggregation are principally responsible for this capability (i.e., anticoagulant activity) [99,100]. Other carrageenan bioactivities, such as anti-tumor, anti-viral, and immunomodulation, have been proven and are commercially exploited [101,102]. In the case of herpes simplex virus types 1 and 2, HIV-1 and human rhinovirus, the carrageenan antiviral system operates by keeping virus particles far from the cell, which is considered to be an encouraging outcome [103,104].

Furthermore, the inclusion of this red seaweed species in the human diet contributes as a flavor enhancer in soups, salads, or seafood, imparting a pleasant mushroom scent [105]. However, the existence of some harmful volatile chemicals, such as bromoform, implies that the inclusion of this fresh seaweed into the human diet should be done with caution [105,106]. Although the application of the *Asparagopsis* sp. to feed cattle with the objective to reduce the methane, the bromoform molecules are a main target to understand their behavior during the seaweed processing [107]. Furthermore, it is supported that the seaweed drying and other type of processing (such as washing or cooking) reduce the bromoform and other volatile (such as iodine) compounds concentration in the seaweeds, mainly the solar drying and seaweed dehydration [108,109,110,111]. Only, the freeze drying maintain the same level of bromoform compounds in the dried seaweeds [108].

## 4. Materials and Methods

### 4.1. Seaweed Harvesting and Preparation

On 27 May 2020, the red seaweed *Chondracanthus teedei* var. *lusitanicus* was harvested in the Portuguese seashore of Buarcos Bay at Figueira da Foz (40.165867, −8.885556). Afterwards seaweeds were placed in plastic bags inside a coolbox and transported to the laboratory where were frozen at −20 °C for prior use. Some days later, the seaweeds were washed with filtered seawater to remove sand, epiphytes and other detritus from the seaweed biomass. Afterwards, the seaweed biomass was washed with distilled water to remove the salt content from the seawater, placed in plastic trays, and dried in an air-forced oven (Raypa DAF-135, R. Espinar S.L., Barcelona, Spain) at 60 °C during 48 h. Dried biological samples were milled (<1 cm) with a commercial grinder (Taurus aromatic, Oliana, Spain) and stored in sterile flasks in a dark and dry place (54% humidity) at room temperature (23 °C).

### 4.2. Mineral and Trace Element Characterization

With the ashes obtained, the mineral content was analyzed through dry mineralization and assessed using flame atomic absorption spectrometry (PerkinElmer PinAAcle 900 T, Waltham, MA, USA) [112]. Phosphorus analysis was performed by spectrophotometry (PG instruments T80+ UV/VIS spectrophotometer, Leicestershire, United Kingdom) [113].

For this analysis, we performed an acid digestion with nitric acid 65% (*m*/*v*), in a water bath at 100 °C around 30 min. Finally, the samples were filtrated for a volumetric flask and the volume adjusted with distilled water. After the necessary dilutions (1:10, 1:100 and 1:500) the analysis was carried out on the atomic absorption spectrophotometer equipped with the cathode corresponding to each element.

### 4.3. Nutritional Profile

#### 4.3.1. Moisture and Ashes Content

According to the international standard method 930.04 of Official Methods of Analysis of AOAC International [114], the moisture content was assessed through the fresh weighting of the algal samples and, then, after oven-drying (Memmert, Büchenbach, Germany) at 60 °C during 48 h. Afterwards, the samples were milled (<1 mm) and, approximately, 2 g of each sample was placed in crucibles and dried at 105 °C for 2 h. Then, the samples were placed in a desiccator until constant weight for being again weighted, in order to calculate the moisture content. In accordance with the AOAC method 930.05, the dried samples at 105 °C were placed in an incineration muffle during 2 h at 550 °C (Induzir, Batalha, Portugal) and further cooled in a desiccator and weighted to assess the ashes amount.

#### 4.3.2. Crude Lipids

The total lipids content was gravimetrically quantified following a continuous ex-traction process with diethyl ether in a Soxhlet apparatus (Behr Labor-Technik GmbH, Düsseldorf, Germany), as it follows the international standard AOAC method 930.09 [114]. The distillation flasks were previously dried at 105 °C for 2 h, cooled in a desiccator and weighted in an analytical scale (Sartorix, Göttingen, Germany). Afterwards, the distillation flasks were filled (2/3 of their capacity) with diethyl ether (Panreac, Chicago, IL, USA). Then, approximately 2 g of the algal samples were packed in filter paper and placed into the thimble. After 16 h of extraction, all the solvent was collected and evaporated (BÜCHI Labortechnik AG, Flawil, Switzerland). The distillation flasks were then dried at 105 °C for 2 h and weighted when cooled down.

#### 4.3.3. Total Nitrogen/Protein

The total nitrogen/protein content was determined by Kjeldhal method (AOAC method 978.04) [114], while it was used 5 as a protein conversion factor [115]. In a Kjeldhal tube, was added approximately 0.5 g of the previously dried algal sample, and then it was added a selenium catalyst (PanReac AppliChem, Darmstadt, Germany) and 12 mL of sulfuric acid (Chem-Lab NV, Zedelgem, Belgium). The tubes were then placed into the Kjeldhal digester (VELP Scientifica, Usmate Velate MB, Italy) at 400 °C for 2 h. The samples could cool in the fume cupboard, and it was added 50 mL of distilled water in each tube and putted into the Kjeldhal distiller. Concurrently, it was placed 30 mL of boric acid (Chem-Lab NV, Zedelgem, Belgium) in an Erlenmeyer (one for sample), being further placed into the Kjeldhal distiller as well (VELP Scientifica, Usmate Velate MB, Italy). To the Kjeldhal tube was added 50 mL of distilled water and 50 mL of sodium hydroxide (NaOH) at 40% (*m*/*v*) (JMGS—José Manuel Gomes dos Santos, Odivelas, Portugal). The distilled solution was collected and titrated with chloridric acid (HCl) 0.1 M (Chem-Lab NV, Zedelgem, Belgium).

#### 4.3.4. Crude Fiber and Total Carbohydrates/Nitrogen-Free Extractives

According to the standard method 930.10 of AOAC [114], the crude fiber was analyzed through the weighting of 2 g from the algal samples, previously oven dried (Memmert, Germany) at 105 °C for 2 h and placed in a 600 mL goblet. It was then added 200 mL of sulfuric acid (H_2_SO_4_) 12.5 g/L (Chem-Lab NV, Belgium) and the samples were placed in a fiber analyzer (Labconco Corporation, Kansas City, MO, USA) for 30 min. After this procedure, the samples were filtered with a filter crucible G2 under vacuum (General Electric, Boston, MA, USA). The residue was then placed into the goblet with 250 mL of sodium hydroxide (NaOH) 12.5 g/L (JMGS—José Manuel Gomes dos Santos, Odivelas, Portugal) and set into the fiber analyzer for more 30 min. With the same filter crucible G2, the samples were again vacuum filtered and dried at 130 °C for 2 h. After the samples were cooled down in a desiccator, they were weighted in an analytical scale (Sartorix, Germany) and placed into an incineration muffle at 550 °C (Induzir, Batalha, Portugal) for 2 h. Finally, the samples could cool down and were weighted to calculate the crude fiber. Nitrogen-free extractives are the difference for 100 of the remaining constituents (moisture, lipids, protein, crude fiber and ash), while the total carbohydrates, corresponds approximately to the difference between 100 and the sum of the moisture, ash, lipids and protein.

### 4.4. Polyssacharides Characterization

Due to carrageenan type variation, the red seaweed *C. teedei* var. *lusitanicus* was separated into phases based on their life cycle using a binocular magnifying glass (Kern & Sohn GmbH, Balingen, Germany).

#### 4.4.1. Carrageenan Extraction

The extraction of carrageenan was carried out in accordance with the method reported by Pereira and van de Velde (2011) [116]. Before extraction, the milled seaweed (1 g) was pre-treated with an acetone (Fisher Chemicals, Portugal): methanol (VWR Prolabo Chemical, Portugal) (1:1) solution in a final concentration of 1% (*m*/*v*) (final volume: 100 mL; 50 mL acetone: 50 mL methanol) for 16 h, at 4 °C, to remove the organic-soluble fraction. The liquid solution was decanted, and the seaweed residues obtained were dried in an air-forced oven (Raypa DAF-135, R. Espinar S.L., Barcelona, Spain) at 60 °C before the extraction.

The dried samples were placed in 150 mL of NaOH (Applichem Panreac, Chicago, IL, USA) (1 M) (1 g of initial seaweed: 150 mL of NaOH solution) in a hot water bath system (GFL 1003, GFL, Burgwedel, Germany), at 85–90 °C, for 3 h. The solutions were hot filtered, under vacuum (Laborport N820, Lisbon, Portugal) through a cloth filter supported in a Buchner funnel. After that, the extract was again filtered under vacuum with a Goosh 2 silica funnel. The extract was evaporated (rotary evaporator model: 2,600,000, Witeg, Germany) under vacuum to one-third of the initial volume (50 mL). The carrageenan was precipitated by adding twice its volume of 96% ethanol (José Manuel Gomes dos Santos, Portugal) (100 mL). The carrageenan precipitated was washed with ethanol 96%, 48 h at 4 °C before dry in an air force oven (60 °C, 48 h) (Raypa DAF-135, R. Espinar S.L., Barcelona, Spain).

#### 4.4.2. FTIR-ATR Characterization

The Fourier Transform Infrared Spectroscopy—Attenuated Total Reflection (FTIR-ATR) examination is a methodology of infrared spectroscopy that is frequently used to investigate and characterize carbohydrates found in seaweeds (among other chemicals) and is based on the procedure outlined by Pereira, Gheda and Ribeiro-Claro (2013) [30].

The dried polysaccharide samples from the previous polysaccharide extraction phases were powdered using a commercial mill and subjected to direct examination without further preparation for FTIR-ATR analysis. This technique requires only a dried milled (<1 mm) sample to be evaluated.

FTIR-ATR spectra were recorded on an Perkin Elmer Spectrum 400 spectrometer (Waltham, MA, USA), with no need for sample preparation, since these assays only required dried samples [31]. All spectra are the average of two independent measurements from 650 to 1500 cm^−1^ with 128 scans, each at a resolution of 2 cm^−1^.

#### 4.4.3. ^1^H-NMR Characterization

^1^H-NMR spectra were made on a Bruker AMX600 spectrometer operating at 500.13 MHz at 65 °C. Typically, 64 scans were recorded with an interpulse delay of 5 s (T1 values for the resonance of the anomeric protons of κ- and ι-carrageenan are shorter than 1.5 s). Sample preparation for the ^1^H-NMR experiments involved dissolving the carrageenan sample (5 mg mL^−1^) at 80 °C in D2O containing 1 mMTSP (3-(trimethylsilyl) propionic-2,2,3,3-d4 acid sodium salt) and 20 mM Na_2_HPO_4_, followed by sonication (Branson 2510) for three periods of 1 h. Chemical shifts (δ) are referred to internal TSP standard (δ = 0 ppm for 1H) according to Knutsen and Grasdalen (1987) [33]. The chemical shift data described by Van de Velde et al. (2002a) [34] were used to assign the ^1^H-NMR spectra.

#### 4.4.4. ^13^C-NMR Characterization

Female and male gametophytes alkaline-extracted carrageenans ^13^C-NMR spectra were obtained on a Varian Unity 500 spectrometer at 125.69 MHz. Samples (15/20 mg mL^−1^) were dissolved in D_2_O and the spectra were recorded at 80 °C, 10.000 accumulations, pulse 15 μs, acquisition time 3 s and relaxation delay 5 s. ^13^C-NMR spectra of tetrasporic carrageenans were recorded on a Bruker AMX500 spectrometer operating at 125.76 MHz, as described in the literature [34,35].

The sample preparation was as follows: a solution of 5 mg mL^−1^ carrageenan in H_2_O was prepared at 80 °C. This solution was sonicated for three periods of 30 min in melting ice (Heat Systems XL 2020 sonicator, 12 mm tip, power 475 W, frequency 20 kHz); the solution was centrifuged at elevated temperature to remove insoluble material. The sonicated material was dialyzed against phosphate buffer (20 mM Na_2_HPO_4_; 3 times 2 L), water (1 time 2 L) and lyophilized. A concentration of 70–100 mg mL^−1^ in D_2_O containing 20 mM Na_2_HPO_4_ and 30 mM TSP, was used to dissolve the material. Chemical shifts (δ) were referred to an external TSP/DMSO standard (δDMSO = 39.45 ppm for ^13^C), in accordance with Usov et al. (1980) [36]. The chemical shift data summarized by Van de Velde et al. (2002) [34] was used to assign the ^13^C-NMR spectra.

## 5. Conclusions

The chemical analysis of the red seaweed *Chondracanthus teedei* var. *lusitanicus* shows that this species has a favorable nutritional profile for human diet and may have some nutraceutical benefits in lowering several common fast food-related disorders (such as diabetes, obesity, or cardiovascular diseases).

Further research is needed on this seaweed, mainly to determine which growing techniques are more adequate to cultivate, and, in such conditions, to study the expected changes in its nutritional profile.

Finally, this red seaweed has nutraceutical potential that should be exploited as a food supplement, which even in low doses can improve human welfare.

On the other hand, this seaweed intake overdosage can have a negative influence on human health, mostly through an uncontrolled mineral intake and can have an impact on cell homeostasis. For this reason, to consider any seaweed proper for human consumption, its nutritional composition must be monitored regularly.

## Figures and Tables

**Figure 1 marinedrugs-19-00570-f001:**
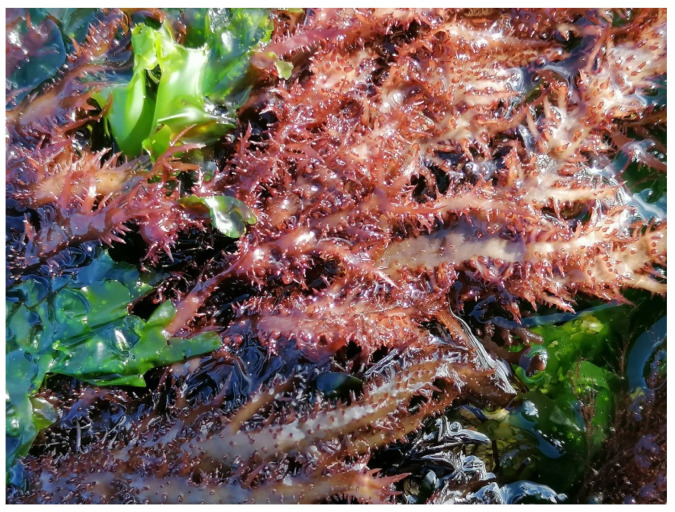
*Chondrachantus teedei* var. *lusitanicus* (fructified female gametophyte) in Buarcos Bay (Figueira da Foz, Portugal).

**Figure 2 marinedrugs-19-00570-f002:**
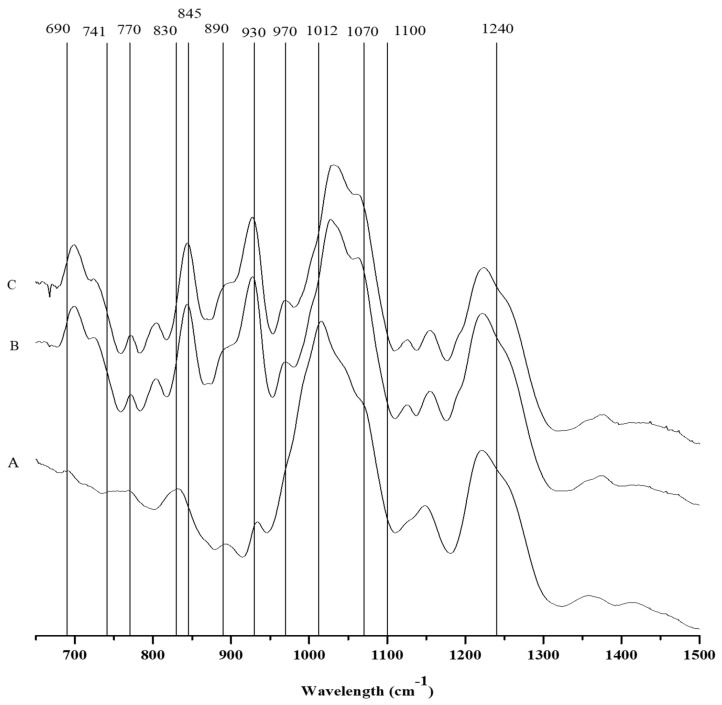
FTIR-ATR spectra of the carrageenophytes: **A**: *Chondracanthus teedei* var. *lusitanicus* tetrasporophyte, **B**: *C. teedei* var. *lusitanicus* male and **C**: *C. teedei* var. *lusitanicus* female gametophytes.

**Figure 3 marinedrugs-19-00570-f003:**
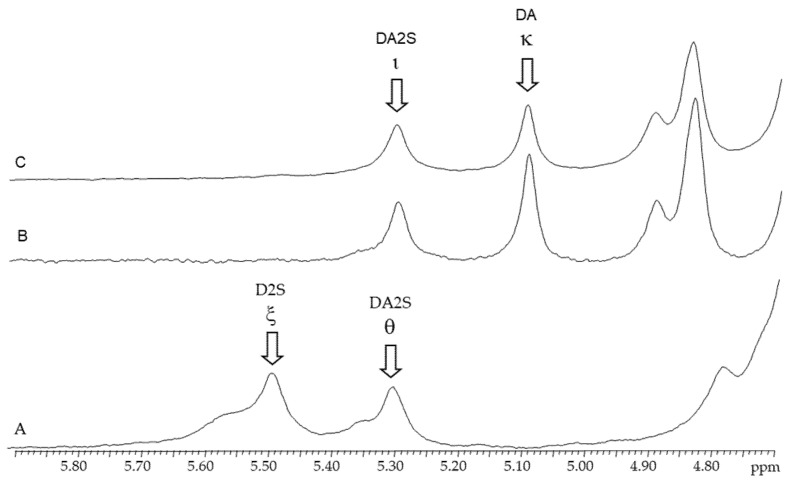
^1^H-NMR spectra of carrageenans extracted from *Chondracanthus teedei* var. *lusitanicus*: **A** (tetrasporophyte). **B** (female gametophyte), **C** (male gametophyte).

**Figure 4 marinedrugs-19-00570-f004:**
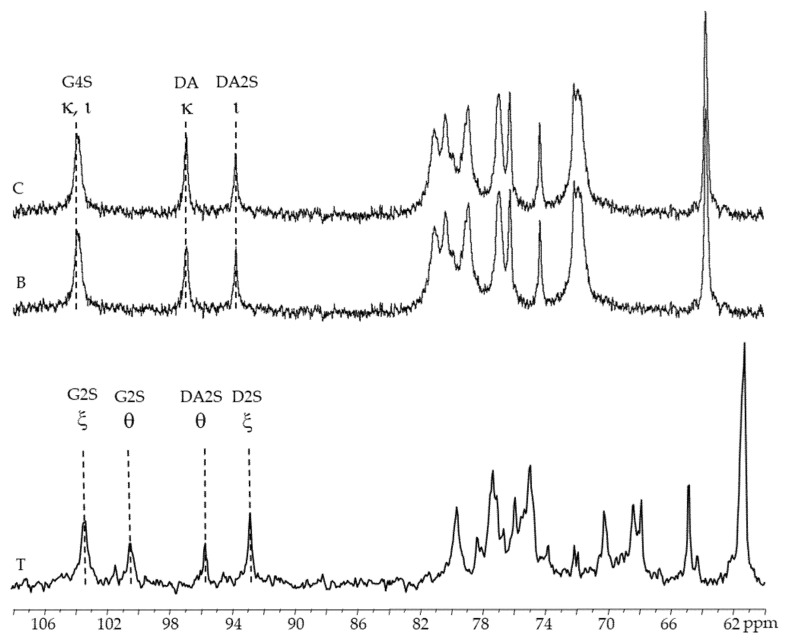
^13^C-NMR spectra of carrageenans extracted from *Chondracanthus teedei* var. *lusitanicus*: **T** (tetrasporophyte). **B** (male gametophyte), **C** (female gametophyte—alkaline extraction).

**Table 1 marinedrugs-19-00570-t001:** Macro- and micro-elements characterization of the dried biomass of *C. teedei* var. *lusitanicus*. Results are expressed in mean ± standard deviation. Nutritional value in 7 g of *C*. *teedei* var. *lusitanicus* according to the established Dietary Reference Intake (DRI) [1,27]. NA—Non applicable.

Macro- and Micro-Elements	g 100 g^−1^	7 g of *C. teedei* var. *lusitanicus* (g)	DRI (%)
Nitrogen	2.13 ±0.01	NA	NA
Phosphorus	0.20 ± 0.01	1.4 × 10^−2^	2
Calcium	0.26 ± 0.03	1.82 × 10^−2^	2.27
Magnesium	0.86 ± 0.02	6.02 × 10^−2^	16.05
Potassium	2.29 ± 0.07	1.60 × 10^−1^	8.02
Iron	0.02 ± 0.03	1.28 × 10^−3^	9.15
Copper	3.0 × 10^−4^ ± 3.0 × 10^−5^	2.33 × 10^−5^	2.33
Zinc	2.4 × 10^−3^ ± 1.0 × 10^−4^	1.68 × 10^−4^	1.68
Manganese	1.2 × 10^−3^ ± 1.0 × 10^−5^	8.40 × 10^−5^	4.20

**Table 2 marinedrugs-19-00570-t002:** Nutritional characterization of the fresh (FW) and dried (DW) *C. teedei* var. *lusitanicus* biomass weight. Results are expressed in mean ± standard deviation. Nutritional value in 7 g of *C*. *teedei* var. *lusitanicus* according to the established Dietary Reference Intake (DRI) [1,27]. NA—Non applicable.

	*C. teedei* var. *lusitanicus* (g 100 g^−1^)			7 g of *C. teedei* var. *lusitanicus* (g)	DRI (%)
	**FW**	**DW**	**DW:FW**		
Moisture	86.52 ± 0.18	NA	NA	NA	NA
Ash	3.96 ± 0.07	29.35 ± 0.13	1:7.4	2.1	NA
Total lipid	0.19 ± 0.01	1.42 ± 0.01	1:7.4	0.10	0.14
Fiber	0.24 ± 0.01	1.78 ± 0.09	1:7.4	0.13	0.49
Protein	1.54 ± 0.01	11.42 ± 0.01	1:7.4	0.80	1.59
Total carbohydrate	7.55 ± 0.12	56.03 ± 0.05	1:7.4	3.98	1.50
Energy (Kcal 100 g^−1^)	38 ± 0.47	283 ± 0.23	1:7.4	19.81 *	0.99

* Measured in total Kcal.

**Table 3 marinedrugs-19-00570-t003:** Polysaccharide quantification and nutritional value in 7 g of *C*. *teedei* var. *lusitanicus* according to the established Dietary Reference Intake (DRI) [1,27].

*Chondracanthus teedei* var. *lusitanicus* Life Cycle Phase	DW (%)	7 g of *C. teedei* var. *lusitanicus* (g)	DRI (%)
Female gametophyte	40.9 ± 1.5	2.86	11.45
Male gametophyte	42.1 ± 4.5	2.95	11.79
Tetrasporophyte	28.1 ± 8.1	1.97	7.87

**Table 4 marinedrugs-19-00570-t004:** FTIR-ATR bands identification and characterization of the *C*. *teedei* var *lusitanicus* (CTGF-female gametophyte, CTGM- male gametophyte, CTT- tetrasporophyte), based on the literature [30,31].

Wave Number (cm^−1^)	Bound	Compound	CTGF	CTGM	CTT
**805**	C–O–SO_3_ on C_2_ of 3,6-anhydrogalactose	DA2S	+	+	-
**825–830**	C–O–SO_3_ on C_2_ of galactose	G/D2S	-	-	+
**845**	D-galactose-4-sulfate	G4S	+	+	-
**867**	C–O–SO_3_ on C_6_ of galactose	G/D6S	+	+	-
**890–900**	Unsulfated b-d-galactose	G/D	+	sh	sh
**905**	C–O–SO_3_ on C_2_ of 3,6-anhydrogalactose	DA2S	sh	sh	sh
**930**	C–O of 3,6-anhydrogalactose (agar/carrageenan)	(DA)	+	+	sh
**970–975**	Galactose	G/D	+	+	-
**1012**	Sulfated esters	S=O	+	+	+
**1070**	C–O of 3,6-anhydrogalactose	DA	+	+	sh
**1100**	Sulfated esters	S=O	+	+	+
**1240–1260**	Sulfated esters	S=O	+	+	+

Sh—shoulder (where the peak shows intensity but not enough to be designated a peak because of the surrounding peak intensities).

## Data Availability

Not applicable.

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
