# Peer review of "Chondracanthus teedei var. lusitanicus: The Nutraceutical Potential of an Unexploited Marine Resource"

_marinedrugs, 2021, doi:10.3390/md19100570_

Round 1

Reviewer 1 Report

The idea of the study entitled "Chondracanthus teedei var. lusitanicus: the nutritional potential of an unexploited marine resource“ is interesting. According to my opinion the paper is simple, well written and the authors follow a logical sequence to discuss their findings.

The introduction leaves room for improvement. Authors should address this section to findings of others on the investigations used in this study on red algae, rather than more general view. Same for the discussion part. There are too many references for this kind of study (simple).

The subject of the paper is "relatively" novel, but there are previous researches on this seaweed species (reported in References 11/12/13/15/17/29/56, etc.) of which some investigated parameters analysed in this study (e.g. ions, soluble proteins, free amino acids and nucleotides, etc.).

The methodology is correct but macro- and micro-elements characterizations by AAS and nutritional profiles that includes only main subgroups of chemical compounds are too simple for scientific study in this high-quality journal.

This paper could be significantly improved by adding other experiments on bioactive chemical components that are present in Chondracanthus teedei var. Lusitanicus like pigments, polysaccharides, polyphenolic compounds, free fatty acids, VOCs, sterols, etc. detected using some specific techniques as chromatography. Also, some biological activity of the fractions could be tested.

The overall quality of the paper is below criteria and requirements of the Marine Drugs journal (IF 5.118).

Author Response

Reviewer 1:

Comment 1: The idea of the study entitled "Chondracanthus teedei var. lusitanicus: the nutritional potential of an unexploited marine resource“ is interesting. According to my opinion the paper is simple, well written and the authors follow a logical sequence to discuss their findings.

Answer 1: The authors acknowledge the kind words from the reviewer.

Comment 2: The introduction leaves room for improvement. Authors should address this section to findings of others on the investigations used in this study on red algae, rather than more general view. Same for the discussion part. There are too many references for this kind of study (simple).

Answer 2: In the introduction it is provided the importance of red seaweeds on human daily diet. Still, for the species Chondracanthus teedei var. lusitanicus, in particular, no studies were found regarding its mineral and element composition.

It was also added in the discussion more information regarding the characterization of this red seaweed polysaccharides and its relevance for human health.

Comment 3: The subject of the paper is "relatively" novel, but there are previous researches on this seaweed species (reported in References 11/12/13/15/17/29/56, etc.) of which some investigated parameters analysed in this study (e.g. ions, soluble proteins, free amino acids and nucleotides, etc.).

Answer 3: Other research in the same genus have been conducted, but to the best of our knowledge, the variety lusitanicus of the species Chondracanthus teedei has not been defined in terms of nutritional value.

Comment 4: The methodology is correct but macro- and micro-elements characterizations by AAS and nutritional profiles that includes only main subgroups of chemical compounds are too simple for scientific study in this high-quality journal.

This paper could be significantly improved by adding other experiments on bioactive chemical components that are present in Chondracanthus teedei var. Lusitanicus like pigments, polysaccharides, polyphenolic compounds, free fatty acids, VOCs, sterols, etc. detected using some specific techniques as chromatography. Also, some biological activity of the fractions could be tested.

The overall quality of the paper is below criteria and requirements of the Marine Drugs journal (IF 5.118).

Answer 4: The authors added more information regarding the polysaccharide characterization of Chondracanthus teedei var. lusitanicus, through FTIR-ATR (Fourier transform infrared spectroscopy – attenuated total reflectance) and nuclear magnetic resonance (NMR) in the different phases of its life cycle.

Reviewer 2 Report

I have carefully reviewed the article. It is well written, especially the discussion of the results. The main concern is that for a research article, for a high impact index journal, such as Marine Drugs, only routine analyzes have been carried out regarding the composition of algae. Concretely:

*Macro- and micro-elements characterization of the dried biomass of C. teedei var. lusitanicus. R

* Nutritional characterization of the fresh (FW) and dried (DW) C. teedei var. lusitanicus biomass weight. (Moisture, ash, total lipid, fiber, protein and total carbohydrate energy).

Thus only two tables with the results are presented.

Check L201-202 where a reference to a Table 3 is done (table 3 is missing).

"As it shown on Table 1 and 2, this seaweed is also a rich supply of microminerals: iron, manganese, copper and zinc, as well as dietary fibers and protein (Table 3)..."

Author Response

Reviewer 2:

Comment 1: I have carefully reviewed the article. It is well written, especially the discussion of the results. The main concern is that for a research article, for a high impact index journal, such as Marine Drugs, only routine analyzes have been carried out regarding the composition of algae. Concretely:

*Macro- and micro-elements characterization of the dried biomass of C. teedei var. lusitanicus. R

* Nutritional characterization of the fresh (FW) and dried (DW) C. teedei var. lusitanicus biomass weight. (Moisture, ash, total lipid, fiber, protein and total carbohydrate energy).

Thus only two tables with the results are presented.

Answer 1: The authors added more information regarding the polysaccharide characterization of Chondracanthus teedei var. lusitanicus, through FTIR-ATR (Fourier transform infrared spectroscopy – attenuated total reflectance) and nuclear magnetic resonance (NMR) in the different phases of its life cycle.

Comment 2: Check L201-202 where a reference to a Table 3 is done (table 3 is missing).

"As it shown on Table 1 and 2, this seaweed is also a rich supply of microminerals: iron, manganese, copper and zinc, as well as dietary fibers and protein (Table 3)..."

Answer 2: Thank you for your warning, the authors corrected the information.

Round 2

Reviewer 1 Report

According to my opinion after incorporating the analysis and the results for polyssacharides from C. teedei var. lusitanicus and with other modifications that have been performed, the manuscript is significantly improved and now I found it suitable for publication in Marine Drugs journal.

Reviewer 2 Report

Dear Authors,

The MS was improved, from my point of view ready to be published.